# Valorization of β-Chitin Extraction Byproduct from Cuttlefish Bone and Its Application in Food Wastewater Treatment

**DOI:** 10.3390/ma15082803

**Published:** 2022-04-11

**Authors:** Nisrine Nouj, Naima Hafid, Noureddine El Alem, Ingrid Ioana Buciscanu, Stelian Sergiu Maier, Petrisor Samoila, Gabriela Soreanu, Igor Cretescu, Catalina Daniela Stan

**Affiliations:** 1Material and Environmental Laboratory, Department of Chemistry, Faculty of Sciences, IBN ZOHR University, Agadir 80000, Morocco; hafidnaima@yahoo.fr (N.H.); n.elalem@uiz.ac.ma (N.E.A.); 2Department of Chemical Engineering in Textiles and Leather, Faculty of Industrial Design and Business Management, “Gheorghe Asachi” Technical University of Iasi, 700050 Iasi, Romania; ibuciscanu@yahoo.com (I.I.B.); smaier@tuiasi.ro (S.S.M.); 3Laboratory of Inorganic Polymers, “Petru Poni” Institute of Macromolecular Chemistry, 41A Aleea Grigore Ghica Vodӑ, 700487 Iasi, Romania; samoila.petrisor@icmpp.ro; 4Department of Environmental Engineering and Management, Faculty of Chemical Engineering and Environmental Protection, “Gheorghe Asachi” Technical University of Iasi, 700050 Iasi, Romania; gsor@tuiasi.ro; 5Department of Drug Industry and Pharmaceutical Biotechnology, “Grigore T. Popa” University of Medicine and Pharmacy, 16 University St., 700115 Iasi, Romania; catalinastan68@yahoo.com

**Keywords:** cuttlefish bone, β-chitin, fish byproducts, industrial, biocoagulant, effluents, Box–Behnken experimental design

## Abstract

The nontoxicity, worldwide availability and low production cost of cuttlefish bone products qualify them an excellent biocoagulant to treat food industry wastewater. In this study, cuttlefish bone liquid waste from the deproteinization step was used as a biocoagulant to treat food industry wastewater. This work concerns a waste that has never before been investigated. The objectives of this work were: the recovery of waste resulting from cuttlefish bone deproteinization, the replacementof chemical coagulants with natural ones to preserve the environment, and the enhancement ofthe value of fishery byproducts. A quantitative characterization of the industrial effluents of a Moroccan food processing plant was performed. The physicochemical properties of the raw cuttlefish bone powder and the deproteinization liquid extract were determined using specific analysis techniques: SEM/EDX, FTIR, XRD and ^1^H-NMR. The protein content of the deproteinization liquid was determined by OPA fluorescent assay. The zeta potential of the liquid extract was also determined. The obtained analytical results showed that the deproteinization liquid waste contained an adequate amount of soluble chitin fractions that could be used in food wastewater treatment. The effects of the coagulant dose and pH on the food industrial effluents were studied to confirm the effectiveness of the deproteinization liquid extract. Under optimal conditions, the coagulant showed satisfactory results. Process optimization was performed using the Box–Behnken design and response surface methodology. Thus, the optimal removal efficiencies predicted using this model for turbidity (99.68%), BOD_5_ (97.76%), and COD (82.92%) were obtained at a dosage of 8 mL biocoagulant in 0.5 L of food processing wastewater at an alkaline pH of 11.

## 1. Introduction

Increasing industrialization has led to the huge production of and a limited capacity for treatment of wastewater. Compared to other industrial sectors, the food industry uses a much larger amount of water for each ton of product [1]. The food processing industries benefit from fishery resources, but significant daily quantities of liquid and solid wastes from the production process need to be treated in accordance with appropriate depolluting methods [2]. Among the less reusable wastes generated in this industrial sector are the remains of exoskeletons, heads, bones and other usually inedible parts from processed marine resources, which are generally drawn and disposed of in conjunction with liquid discharges during cleaning processes [3].

The treatment of food wastewater (FW) has been a problem of major environmental concern. However, processes such as coagulation and flocculation have been found to be straightforward and cost effective [4,5,6]. The use of conventional coagulation/flocculation agents is frequently subject to discussion, due to changes in environmental regulations [7,8]. This issue has led to the search for another alternative to preserve the environment and reduce health problems. The process of biocoagulation would be more environmentally friendly than other methods, and it would be useful as an alternative to minimize these risks [9]. Many types of biocoagulants have been developed to remove pollutants from wastewater, including: chitosan [10], tannins [11], aqueous extract from moringa oleifera seed [12], plantain peel ash extract [13] and nirmali seeds [14]. These coagulants have the advantage of being biodegradable and without risk to public health, unlike other materials based on plants, animals or micro-organisms used in wastewater treatment [15]. A marine biocoagulant involves the extraction of the abundant biopolymer chitin and its derivative chitosan. As part of the extraction of the above-mentioned compounds, a huge quantity of waste is obtained [16]. The deproteinization step generates a liquid that is usually evacuated and needs to be well managed [16].

Cuttlefish bone (CB) is a highly porous hard tissue of cuttlefish, which functions as a rigid floating reservoir inside the animal body [17]. It is an inorganic-organic composite material made up of aragonite, β-chitin and protein [17], from which the biopolymer β-chitin is extracted for several uses [18]. Compared with α -chitin, it shows higher solubility and swelling capacity, due to much weaker intermolecular hydrogen bonding attributable to the parallel arrangement of the main chains. Therefore, it can be expected to have a higher affinity for various solvents and to be more reactive than α-chitin [19]. Recently, β-chitin has been shown to exhibit higher reactivity than α-chitin in the deacetylation process and to have a high availability to participate in many chemical transformations [20].

Cuttlefish bone is used in many fields, especially for toothpaste production [21]. It is generally used as an adsorbent for heavy metals removal from water. Recent studies showed its effectiveness for removing significant pollutants such as fluoride from drinking water [22], dyes from textile wet processing wastewater [23] and Cobalt (II) from aqueous solution [24].

The liquid resulted from CB deproteinization had never been studied to evaluate its chitin content. In this work, the liquid extract (LC) resulted from the deproteinization of cuttlefish bone was studied and valorized as a biocoagulant for food wastewater treatment. The effect of coagulation treatment parameters (initial pH, biocoagulant dosage, temperature) upon the efficiency of wastewater indicators reduction (turbidity, biochemical oxygen demand (BOD_5_), chemical oxygen demand (COD)) was studied using the Box―Behnken experimental design.

## 2. Materials and Methods

This study was conducted using experimental and modeling approaches. The experimental step was divided in two parts: (1) performing the deproteinization of the *Sepia officinalis* and separating the liquid extract, and (2) treating the FW using β-chitin liquid extract. A Box–Behnken experimental design and response surface methodology (RSM) were used for modeling and optimizing the coagulation treatment of wastewater.

### 2.1. Collection of Wastewater and Cuttlefish Bone

Food wastewater samples were collected from a fish processing plant located in the industrial district of Anza, Agadir city, Morocco, exhibiting similar characteristics to those mentioned in our previous work [25]. The main objective was to test different biocoagulants for similar effluents in order to point out their effectiveness for wastewater treatment, with the possibility of reusing the treated water. Standard methods for the evaluation of water and wastewater (APHA) were used to analyze the physicochemical quality (pH, temperature, turbidity, chemical oxygen demand (COD) and biological oxygen demand during 5 days (BOD_5_)) [26].

The bones of *Sepia officinalis* were collected from a local market in the city of Agadir, Morocco, and washed several times with distilled water to remove unwanted debris (Figure 1a). The clean cuttlefish bone (CB) was dried at 220 °C for 15 min, and then ground with a mortar and pestle to produce a fine powder. The obtained material was sieved to a size of 200 µm and stored in this condition for 7days to remove any residual moisture.

### 2.2. Preparation of the Deproteinization Liquid Extract

A total of 100 g of CB was mixed with 1 L of 1 M NaOH and stirred at 70 °C for 2 h. The mixture resulted from the deproteinization process was allowed to settle and cool down, and then it was separated by filtration. The resulting solid has the potential to be further processed to obtain chitosan; the supernatant mainly contains hydrolyzed proteins and is regarded as a waste [27]. The liquid extract was allowed to settle for 30 min and stored for further use (Figure 1b).

### 2.3. Separation and Purification of Chitin Forms from the Deproteinization Liquid Extract

The separation and purification of chitin fractions from the LC extract was performed as described below. A volume of 100 mL (pH 13.2) was neutralized with 36% HCl to pH 5.8 ÷ 6.0. Then, 70 mL of 96% ethanol were added and the resulting solution was concentrated by azeotropic distillation until the volume decreased to about 50 mL. Another portion of 70 mL ethanol was added to the concentrated solution, and the azeotropic distillation was continued until salt precipitation occurred (a volume of about 8 mL resulted). After cooling at room temperature, the crystallized salts were removed by filtration and washing with 10 mL ethanol. The volume of the resulted solution was adjusted to 20 mL with distilled water (dH_2_O), 5 mL of 0.01 M trichloroacetic acid solution (pH 2.1) were added, and the mixture was heated at 90 °C under reflux for 30 min, in order to precipitate the polypeptides and to maintain polysaccharidic fractions in the solution [28]. The cooled suspension was centrifuged at 1800× *g* for 30 min, and the gelatinous sediment was removed. A total volume of about 22.5 mL alcoholic acidic solution resulted. To remove the trichloroacetic acid and the organic solvent, the alcoholic solution was diluted with dH_2_O to a 1:1 ratio and subjected to dialysis through a Spectra/Por ® 3 (Spectrum Laboratories Inc., Rancho Dominguez, CA, USA) MWCO 3.5 kDa membrane, against 1 L of dH_2_O, changed four times until the pH increased to about 5.3. The resulted suspension that included insolubilized pellets was concentrated by vacuum evaporation to 3 mL and freeze dried. The solid fraction was resolubilized in 10 mL of a mixture of 85% anhydrous methanol and 15% ethylic ether (prior to mixing, both solvents were cooled at 4 ± 1°C), vacuum filtered and evaporated in a vacuum oven at 35 °C. For both operations, a vacuum was produced by a water jet pump to avoid the flammability hazard. The resulting solid was stored in a sealed glass vial.

### 2.4. Scanning Electron Microscopy and Energy-Dispersive X-ray Spectroscopy (SEM/EDX)

The surface morphology of the raw CB powder and purified chitin fractions (CHT) separated from the LC extract was determined using a scanning electron microscope (SEM) (JEOL, Akishima, Tokyo, Japan) Supra 40 Vp Gemini Zeiss Column, with a maximum voltage of 20 kV. The elemental composition of the raw cuttlefish bone sample was also defined by energy dispersive X-ray analysis (EDX).

### 2.5. Fourier Transform Infrared Spectroscopy (FTIR)

Attenuated total reflection Fourier transform infrared spectroscopy (ATR-FTIR, SHIMADZU, IRAffinity-1S, Paris, France) was performed to obtain the FTIR spectra of raw CB, LC extract and CHT fractions separated from the LC extract. The device is equipped with a Jasco ATR PRO ONE module, with a resolution of 16 cm^−1^ at room temperature, in the wavelength range of 4000 cm^−1^ to 400 cm^−1^.

### 2.6. X-ray Diffraction (XRD)

The raw CB powder and chitin forms separated from the LC extract were characterized using an X-ray diffractometer (Bruker D8 Twin, Iéna, Germany) using monochromatic λ(KαCu) =1.5418 Å radiation at 40 kV and 40 mA. The CB and CHT samples were tested in the range 3° < 2θ < 60° at a step size of 0.02° and a scan rate of 0.3 s/step.

### 2.7. Ultraviolet/Visible Spectroscopy (UV-Vis)

The visible ultraviolet spectrum (UV-Vis) was measured by spectrophotometry (U-2910, Hitachi, Santa Clara, CA, USA) in order to analyze the deproteinization of cuttlefish bones. Measurements were made in the range of 200 to 800 nm.

The optical properties of the tested samples were interpreted by studying their absorption of radiation at various wavelengths: 436 nm, 525 nm and 620 nm. All samples were tested in their liquid form.

### 2.8. OPA Fluorescent Assay for Protein Quantification

Protein content in the liquid extract was assessed by the OPA assay procedure. The OPA stock solution was prepared by dissolving 120 mg of o-phthalaldehyde Merck KGaA, Darmstadt, Germany) in methanol and then diluting in 1 M boric acid to 100 mL, followed by pH adjusting to 10.4 with potassium hydroxide. A volume of 0.6 mL of polyoxyethylene lauryl ether was added. The stock solution is stable for 3 weeks at room temperature. The determination was conducted by the established procedure [29]. Protein absorbance was determined using a spectrofluorometer (RF-6000, Shimadzu, Kyoto, Japan) at an excitation wavelength of 340 nm. Fluorescent emission was read in the range of 390–700 nm.

### 2.9. Proton Nuclear Magnetic Resonance (1H-NMR)

Proton NMR spectra of CHT separated from the LC extract were recorded using a Bruker NEO-1 400 (Bruker, Billerica, MA, USA) spectrometer at 25 °C.

### 2.10. Zeta Potential Measurement

The tendency of chitin oligosaccharides to induce coagulation/flocculation was studied by the electrophoretic light scattering (ELS) technique, using a Zetasizer Nano ZS (Malvern Panalytical, Malvern, Worcestershire, UK) device. Purified CHT resulting from the deproteinization liquor through the procedure described above (Section 2.3) was re-dispersed in distilled water. The zeta potential of the resulted colloidal suspension was measured at pH values in the range of 7 to 11.

### 2.11. JarTest

The coagulation tests were performed using a jar-test. The food wastewater was distributed into six 1 L glass beakers and treated with different dosages of biocoagulant (from 1 mL to 12 mL) at different pH values (from 1 to 12). The rapid mixing stage of about 5 min was followed by slow mixing for about 20 min; then, the solution was allowed to settle for 30 min. Turbidity, BOD_5_ and COD were measured on the collected supernatant; the treatment efficiency in terms of percentage removal rate (%) was calculated using the following equation:(1)$$\mathrm{Removal}\;\mathrm{Rate}\;\left[\%\right]=\frac{\left(Ci-Cf\right)}{Ci}\times 100\%\;$$
where: Ci and Cf are the initial and final values of each studied parameter.

### 2.12. Analytical Methods

A 3-factor, 3-level Box–Behnken design was adopted in this study as the experimental design model. This method is preferred as a design model, since relatively few combinations of the variables are required to estimate the potentially complex response function. A second order polynomial model with interaction terms fitted to the experimental data obtained from the experimental tests conducted on the basis of the Box–Behnken design model is given by the equation:(2)$$Y=\alpha_{0}+\overset{k}{\underset{i=1}{\sum }}\alpha_{i}X_{i}+\overset{k-1}{\underset{i=1}{\sum }}\overset{K}{\underset{j=2}{\sum }}\alpha_{\mathrm{ij}}X_{i}X_{j}+\overset{k}{\underset{i=1}{\sum }}\alpha_{\mathrm{ij}}X_{i}^{2}+\varepsilon \;$$
where α_0_, α_i_, α_ii_ and α_ij_ are the regression coefficients for the intercept, linear, quadratic and interaction terms, respectively, X_i_ and X_j_ are the independent variables (experimental parameters), and ε is the error [30].

Nemrod Wsoftware (v. 2000-D, LPRAI, Marseille, France) was used to perform the experimental design, coefficient determination, data analysis and graphical representation. A random order and three central replicates were applied to the model for the calculation of the pure experimental error [31]. The response surface and the desirability function were used to define the design space and find the optimal conditions. In total, 15 experiments are required to calculate 10 coefficients of the second order polynomial equation, which has been fitted to the experimental data (Table 1). Turbidity (Y_1_), BOD_5_ (Y_2_) and COD (Y_3_) were considered to be the responses of the Box–Behnken design. The three chosen parameters: coagulant dose (X_1_), pH (X_2_) and temperature (X_3_) were considered as independent input variables (Table 2).

## 3. Results and Discussion

Treated food wastewater has been characterized in a previously published article [25]. Additionally, COD was determined in order to evaluate the tested biocoagulant removal capacity of this water indicator. According to the data in Table 3, the values of all parameters were above the Moroccan limits for the discharge of wastewater into the environment, except for pH values. 

### 3.1. Characterization of Raw Cuttlefish Bone and Deproteinization Liquid Extract

#### 3.1.1. Analysis by SEM of Cuttlefish Bone and Chitin Fractions from the LC Extract

The SEM images show that the dried CB powder (Figure 2g) contains clear pores on its surface (Figure 2c,e,f,h). The CB has two faces: the lamellae (Figure 2b) and the dorsal shield (Figure 2d). The internal structure of the bone, on the lamella side, reveals thin layers connected internally by a parallel thin wall (or lamellae). This side shows the internal matrix composed of well-spaced lamellae (Figure 2c), enlarged in (Figure 2f). The walls on either side of a lamella are slightly offset, which can provide better mechanical properties to the fenestration [32]. The surface of the lamellar layer has a wavy appearance, with spaced grooves, possibly due to the growth stages along the pillar column [33,34,35]. The dorsal shield (Figure 2d and its SEM image in Figure 2e) is mainly composed of aragonite, the main polymorph of calcium carbonate, and the compound abundant in Mollusca species.

The presence of numerous surface pores corresponds to the huge hole shown on the SEM image (Figure 2f), confirming the excellent porosity of the marine biomaterial.

According to Figure 3, the SEM images show a slightly smoother surface (Figure 3a,b). Significant pores were not observed on the surface. The chitin (in Figure 3c,d) showed a slightly rough morphology. This difference between the CB powder and the chitin extracted from the liquid extract after treatment is due to the lamella broken by the alkaline medium. The slightly smooth and rough shapes depend on the dorsal or lamellar side.

Figure 3 also shows that the CHT fraction separated from the LC extract has a predominantly amorphous structure, even though a partial crystallinity was identified by XRD analysis (Section 3.1.4). High uniformity of CHT, also obvious on SEM micrographs, is characteristic of β-chitin and can be related to the high structural homogeneity of cuttlefish bone, from which β-chitin is extracted.

#### 3.1.2. Analysis by Energy-Dispersive X-ray Spectroscopy (SEM/EDX)

The EDX spectra of CB lamellae and the dorsal shield (Figure 4a,b) show similar elements. The main constituent of the CB powder is calcium carbonate (CaCO_3_), with trace quantities of Al, Na and Cl. The analysis revealed that the thin films that construct the inner structure of the CB are entirely organic, with no presence of the inorganic phase. These lamellas were also enveloped in a thin organic membrane that was identified as β-chitin. This biopolymer is sparse, less stable and less crystalline. In particular, β chitin is more reactive than α chitin, especially towards certain solvents [36,37,38,39]. Cuttlefish bone consists mainly of a complex of calcified pillars organized in different layers, with small amounts of β-chitin and proteins as an organic membrane [40,41]. These structures are made up of hierarchical layers that optimize the components of each organic layer, made up of chitin, proteins and calcium carbonate [42].

Many studies confirmed the presence of calcium carbonate CaCO_3_ as the main element of cuttlefish bone using the EDX analysis [38,40].

#### 3.1.3. Analysis by Infrared Spectroscopy (FTIR)

The FTIR spectrum of the raw CB powder presented in Figure 5 shows bands of aragonite detected at 712 cm^−1^, 851 cm^−1^ and 1083 cm^−1^ [34]. A small contribution of chitin is noticed in the range 1030–1110 cm^−1^ (C=O stretch) [34]. In addition, the bands between 600–700 cm^−1^ may also be due to β-chitin present in the CB powder [43,44]. Chitin is not easily detectable in the investigation, due to its presence in the walls of aragonite [45]. Bands close to 1785 cm^−1^ and 2527 cm^−1^ can also be attributed to the vibrations of carbonate ions. The peak of the order of 2927 cm^−1^ indicates the presence of aliphatic C-H stretching [35]. A broad peak at 3422cm^−1^ was ascribed to the O-H stretching vibration mode of the hydroxyl group [35]. The liquid recovered after deproteinization was studied by FTIR spectroscopy in order to identify its main constituents (Figure 6).

The IR spectrum noted here is similar to those obtained in other studies [26,31,40,44]. The difference between the actual CB IR spectrum and those from other studies may be attributed to different species of cuttlefish investigated.

In the FTIR spectra of the liquid recovered after deproteinization, some new bands were revealed that showed the presence of β-chitin, starting with a typical band at 1603 cm^−1^ [46]. Likewise, the axial C-H bond which appears in the 3000–2800 cm^−1^ range is due to the presence of chitin biopolymer [47]. The specific signals attributed to the axial vibrations of the OH and NH groups are observed at 3700 cm^−1^ and 3000 cm^−1^ [48]. The protein band is visible around 2000 cm^−1^ for the secondary amide bond N-H [44]. The band around 1415 cm^−1^ is attributed to C-H rocking [49]. Peaks in the 2850–2925 cm^−1^ range correspond to a symmetrical and asymmetrical stretch of the -CH_2_ group [50]. An absorption band can appear in the 1320–1210 cm^−1^ range, which could be attributed to C-H stretching [51]. The broadband recorded at about 1020 cm^−1^ corresponds to the skeletal vibration of C-O [47]. The peak at 814 cm^−1^ is assigned to the anomeric region of the carbohydrate, or (C-H) [47], which may also indicate the presence of chitin polysaccharides in LC.

The chitin oligosaccharide fraction separated from the LC extract according to the procedure described in Section 2.3 was also investigated by FT-IR (Figure 7).

From the obtained spectrum, together with the deconvolution of two spectral domains, the following characteristic bands were identified, either individual or convoluted: 896 cm^−1^ for the glucopyranose ring; 952 cm^−1^ and 1028 cm^−1^ correspond to the skeletal vibration of C-O, and 1632 cm^−1^ for amide III, 1732 cm^−1^ for the acetyl–ester bond, 1628 cm^−1^ and 1670 cm^−1^ for amide I, 1558 cm^−1^ for the N–acetyl–ester bond, 2920 cm^−1^ for CH_2_ and 3434 cm^−1^ for the hydroxyl group.

#### 3.1.4. Analysis by XRD Spectroscopy

The X-ray diffraction shows that the cuttlefish bone exhibits well-crystallized forms, with aragonite as the only mineral phase present in the sample (Figure 8a). The most significant peak is at around 30° and corresponds to aragonite, while the other peaks indicate variable elements based on calcium carbonate. Essentially, the spectrum of the cuttlefish bone resembles the diffraction pattern of aragonite to a high extent, and the three most intense reflections are for the (111), (200), and (022) planes. In a diffractogram obtained for a powdered CB sample, the reflections of the main organic component, chitin, are not visible. A broad crystalline reflection in the 25° 2θ range can be attributed to β-chitin, which is the most abundant organic constituent of the *Sepia officinalis* bone [34].

According to Figure 9, the main patterns of CB powder are similar to aragonite patterns, confirming the major presence of this element in the analyzed sample. As confirmed by other studies, calcium carbonate in cuttlefish bone is mostly in the form of aragonite [22,33,51].

The XRD spectrum of the CHT fraction isolated from the deproteinization liquid is represented in Figure 8b. The typical peaks at 2θ of about 8.5, 20 and 28 degrees are obvious, indicating the presence of a small amount of chitin [46].

#### 3.1.5. Ultraviolet/Visible Spectroscopy (UV-Vis)

The UV-Vis spectra of the liquid resulting from CB deproteinization is shown in Figure 10. Analysis was performed at two dilution levels of the initial LC extract: 0.8% and 3%. Both samples showed the highest absorption around 200 nm, which was the shortest wavelength at which absorption was involved. The bands showed/visible in the spectra obtained for the 0.8% solution between 200 and 280 nm are linked to certain chromophores present in proteins [52], or aromatic amino acids [53] capable of absorbing light at wavelengths lower than 300 nm. At the studied concentrations, a band appears at 745 nm, which could be explained by the alkaline medium used to remove proteins from the CB powder [54].

#### 3.1.6. Determination of Protein Content of the LC Extract

According to the OPA assay for protein quantitation, the fluorescent emission of the samples in the range of 440–455 nm is attributed to the protein content, regardless of the complexity of the recorder emission spectra. To accurately quantify the amount of large polypeptides resulting from deproteinization, two fitting algorithms were applied to the fluorescence spectrum of the liquid samples, using the Pseudo-Voigt and log-normal functions, respectively. Figure 11 depicts the spectrum fitting results. The Pseudo-Voigt fitting was used to effectively quantitate the polypeptide amount, while the log-normal fitting was used for confirmatory purposes of the fluorescence emission in the range of 440–455 nm. For both fitting algorithms, the third deconvoluted peak (shown in orange in Figure 11a,b) lies in the appropriate emission range (at 442.5, with an amplitude of 3950.64 a.u., and 449.9 nm, with an amplitude of 1219.89 a.u., respectively). Therefore, the presence of large polypeptides is confirmed by the corresponding peak in Figure 11b, and the amount of 0.407 nM/mL was calculated using the amplitude of the third Pseudo-Voigt peak in Figure 11a. The protein amount was calculated based on a bovine serum albumin (BSA) linear calibration curve with the algebraic form: y = 24,807.443 x − 6160.327. The relatively large amount of polypeptides proven by our experiments/determination could contribute to the formation of the chito-protein complexes, which improves the biocoagulation process.

#### 3.1.7. NMR Analysis of Chitin Oligosaccharides Separated from the LC Extract

The alkaline extract resulting after deproteinization contains large amounts of inorganics, together with polysaccharides of chitin origin and residual polypeptides. In order to remove the inorganic fraction, the neutralized liquid extract was diluted with ethylic alcohol and then concentrated by azeotropic distillation, until salts precipitation [55,56]. The organic fraction in the aqueous alcoholic solution was then treated with trichloroacetic acid, which solubilizes chitin oligosaccharides, but precipitates polypeptides. At the end, the organic fraction was brought to a solid state. The oligosaccharides were purified and freeze-dried, in order to obtain clean NMR spectra, devoid of parasitic organic chemical species. The polysaccharide fractions collected after dialysis and freeze-drying are visible in Figure 12.

The ^1^H-NMR spectra recorded for the oligosaccharides separated from LC were interpreted by comparison with the spectrum of the low molecular weight water-soluble chitosan (LMWSC) in Figure 13 [57] because the latter includes the N-acetylated group.

The enhanced resolution ^1^H-NMR spectrum of the chitin-based oligosaccharides separated from the LC extract is presented in Figure 14.

To demonstrate that chitin oligosaccharides bearing N-acetyl-glucosamine groups were obtained from the LC extract, peak g (a triplet) in Figure 13 should be present in spectra recorded for the studied material, between 1.95 and 2.0 ppm. This peak was found at 1.99 ppm (Figure 14), together with the peaks at 2.08 and 2.12, also related to the N-acetyl-glucosamine moiety [58,59]. The intense peak at 3.01 ppm is assigned to the proton at C2 of N-acetyl-glucosamine.

Mass balance performed throughout the separation steps showed that the concentration of chitin fractions extracted from the raw CB was 32 mg/L deproteinization liquid extract (LC).

#### 3.1.8. Zeta Potential Measurements

Coagulation/flocculation processes benefit from the conditional colloidal stability of the oligo- and macro-molecules (or any particulate entities) in solution or in suspension. Both the target suspension to be coagulated and the coagulant are involved in the coagulation/flocculation process through their physical-chemical characteristics. Commonly, the parameter that switches between the stable and unstable state of a colloidal system is its pH value, while repulsion between dispersed particles, expressed by zetapotential, is a measure of their mutual stability. The coagulation/flocculation process occurs when the surface of the particulate entities in suspension reaches low zeta potential values. In the case of organic matter in aqueous suspension, zeta potential values that predispose to coagulation/flocculation are in the range from −8 mV to +3 mV (or, for complex systems like wastewaters, from −5 mV to + 5 mV), while the range of general colloidal instability extends from −30 mV to + 30 mV [60]. The coalescence threshold of emulsions is placed between zetapotentials of −11 mV to −20 mV [61]. Zeta potential values of chitin oligosaccharides separated from the CB deproteinization liquid extract, measured at pH values between 7 and 11, are given in Table 4.

According to the reported data, chitin oligosaccharides in aqueous suspension are subjected to colloidal instability at neutral and alkaline pH; hence, they can act as coagulants/flocculants at mild alkaline pH values and are able to induce extensive disruption of colloidal stability at higher alkaline values.

### 3.2. Effect of pH on the Coagulation Treatment Efficiency

The influence of pH on the removal rate of turbidity, BOD_5_ and COD was evaluated over a 1 to 12 range of pH units, after 30 min of sedimentation.

The main interaction between LC and food wastewater particles is basically by electrostatic attraction. Increasing the pH value, while keeping the other factors constant, leads to the decrease in the values of the measured parameters. The optimum pH for the effective removal of turbidity, BOD_5_ and COD was found to be equal to 11 units; the removal is effective at alkaline pH.

The main mineral component, proven by SEM, EDS, FTIR and XRD characterizations of the CB powder, is aragonite. CB calcium carbonate is identified as an efficient adsorbent by many studies [23,35,44]. CB powder dissolved in water produced [HCO_3_]^−^, [CO_3_]^2−^, [Ca]^2+^, [Ca(HCO_3)_]^+^ and [CaOH]^+^. Several studies [62,63] showed that Ca^2+^ and CO_3_^2−^ were the potential determining ions in a pure calcite solution. The elimination of proteins generates the elimination of a portion of chitin, present in the chito-protein complex. The deproteinization process used in this study falls within the accepted range of treatment methods for obtaining chitin [64]. A portion of chitin will remain in the protein-rich liquid extract, and the main elements responsible for reducing colloids are likely calcium carbonate and β-chitin. However, the drastic conditions of deproteinization may modify the native structure of β-chitin [65].

Effectively decreasing pH provides more Ca^2+^ in a solution obtained from the dissolution of cuttlefish bone powder and increases the number of positive sites. According to the results presented in Figure 15, a higher removal efficiency of turbidity, BOD_5_ and COD from FW is achieved at alkaline pH than at acidic pH. At a pH of about 6, the removal percentage of turbidity and COD was close to the alkaline performance. However, BOD_5_ removal had a lower efficiency, which prompts us to seek better performance, especially for this parameter. Food wastewater contains a large percentage of organic matter that must be removed to avoid harming the environment. That could be the reason behind the changes in the CB behavior after treatment. These changes may be due to the ion exchange reactions, increasing the negative charges at the molecular surface, which increase the sorption phenomenon, leading to a change in the electro-kinetic properties.

### 3.3. Effect of Coagulant Dosage on the Coagulation Treatment Efficiency

The effect of coagulant dosage (from 1 to 12 mL/0.5L) at 30 min sedimentation on the three studied parameters (turbidity, BOD_5_ and COD), taking into account the optimum pH value for FW treatment, are shown in Figure 16. It can be seen that all coagulant dosages performed well at pH = 11 in terms of COD removal; a slight variation for BOD_5_ removal was noticed, while turbidity removal required a dose greater than 4 mL. This may be due to the fact that under-dosage produces destabilized particles because of a greater distance between the particles and a weaker electrostatic interaction. The excess dosage may produce a charge reversal, which results in particle stabilization [66]. The proximity of the ions weakens their activity, although the physical quantity of the dissolved solids is not affected.

Based on the above observations, the flocs produced by LC appear rapidly and can be easily settled, indicating the efficiency of LC in coagulation. Insufficient or under-dosage of the biocoagulantresulted in poor performance in pollution parameter reduction. From these experiments, it can be concluded that the 8 mL dosage of the LC extract shows the optimum efficiency in reducing turbidity (99.68%), BOD_5_ (99.66) and COD (82.92%).

### 3.4. Optimization Using the Box–Behnken Experimental Design

#### 3.4.1. Statistical Analysis

To assess the influence of some key parameters on the efficiency of FW coagulation treatment, three second order quadratic polynomial models were statistically established, using the Box–Behnken design and RSM based on the experimental data. In this regard, the fitted model equations for turbidity, BOD_5_ and COD removal are given in equations (A), (B) and (C), based on ANOVA results (Table 5). According to the sum of the squares of the sequential model, the models were selected based on the highest order polynomials where the additional terms were significant. The experimental results were evaluated and the regression function (Yi) expressed in terms of waste water treatment efficiency for turbidity (A), BOD_5_ (B) and COD (C) and the studied process parameters (Table 5) are given bellow:$$\begin{array}{c} Y_{1}=99.130-0.248\;X_{1}+1.315\;X_{2}+0.273\;X_{3}-0.386\;X_{1}^{2}-2.226\;X_{2}^{2}+0.644\;X_{3}^{2}+1.017\;X_{1}X_{2}+0.097\;X_{1}X_{3}-0.328\;X_{2}X_{3}\\ Y_{2}=98.800-0.189\;X_{1}+13.357\;X_{2}+0.014\;X_{3}-8.011\;X_{1}^{2}-26.569\;X_{2}^{2}+6.634\;X_{3}^{2}+2.260\;X_{1}X_{2}-0.248\;X_{1}X_{3}+5.005\;X_{2}X_{3}\\ Y_{3}=81.940-1.185\;X_{1}+10.085\;X_{2}+0.145\;X_{3}-12.110\;X_{1}^{2}-19.065\;X_{2}^{2}+10.875\;X_{3}^{2}+8.890\;X_{1}X_{2}-0.705\;X_{1}X_{3}+1.285\;X_{2}X_{3} \end{array}$$

The high statistical significance and usefulness of the model equations developed for the studied parameters were confirmed by a low *p*-value (<0.01 ***). In addition, the values of the coefficient of determination (R^2^) (greater than 0.90 for the three parameters) showed that less than 1% of the total variations in turbidity, BOD_5_ and COD removal efficiencies did not coincide with the models developed. The high values of the adjusted R^2^ demonstrate the higher predictive power of our models. The difference between the adjusted R^2^ and predicted R^2^ values is less than 0.008, indicating high consistency between actual responses and statistically predicted responses. The *p* values indicated a good fit of the models developed to the data actually studied (Table 6). Compared to other studies adopting RSM and Box–Behnken statistical analysis for the optimization of industrial wastewater treatment [67,68,69], the proposed models were highly significant and indicated excellent correlations between experimental results and predicted values of COD, BOD_5_ and turbidity removal rates.

#### 3.4.2. Response Surface Plotting and Optimization of Turbidity Removal

Two- and three-dimensional response surface plots were drawn to determine the individual and interactive effects of the variables on the response. Figure 17 clearly shows that the efficiency of turbidity removal increases rapidly with increased pH at coagulant doses not exceeding 10 mL/0.5 L FW. This may be due to the competition of colloids for biocoagulation sites. A higher pH generates more perhydroxyl anions, which promotes the formation of aggregates in the process of biocoagulation [70].

Temperature is considered to be one of the key parameters of the biocoagulation process, although, in the studied correlation between temperature and pH, this parameter slightly affects the variation of turbidity (Figure 18). The interactive effect of pH and temperature results in an optimum treatment temperature of 25 °C at pH 10 and pH 12, respectively. Figure 19 shows 3D and 2D contour plots plot for temperature and coagulant dose, when pH was kept constant at 11 units. The temperature does not affect the dose of the coagulant, which improves the performance of cuttlefish bone as a coagulant.

Subsequently, the maximum of turbidity removal (99.85%) was obtained using 12 mL of biocoagulant at 25 °C at pH 11. In addition, according to ANOVA, pH is an important factor in this study because it affects the overall process of biocoagulation.

#### 3.4.3. Response Surface Plotting and Optimization of BOD_5_ Removal

The combined effect of pH and coagulant dose on the percent removal of BOD_5_ while maintaining constant temperature at the central level is shown in the 3D and 2D contour plots from Figure 20. The increase of BOD_5_ removal efficiency could be due to the increase of the LC extract concentration, which leads to the formation of surface and binding sites in alkaline media. The maximum percentage removal (98.80%) was obtained with a dose of 10 mL coagulant/0.5 L FW at pH 11. The interactive effect of pH and temperature maintaining the dose of coagulant at 10 mL disclosed that the removal of BOD_5_ is sensitive to the change in temperature, taking into account the variation in pH (Figure 21). The reason behind this case could be the optimal conditions of the microbiological flora present in the wastewater. An asymmetric correlation between coagulant dose and temperature is shown in Figure 22, where the pH value is kept at 11.

From the previous graphical representations, it can be seen that the removal efficiency increases under two different conditions: at 10 mL and 12 mL and a temperature set at 21 °C and 25 °C, which may be due to the sensitivity of the microbiological flora.

#### 3.4.4. Response Surface Plotting and Optimization of COD Removal

A strong correlation was recorded between the coagulant dose and the wastewater treatment pH. Once the pH increases, the removal of COD increases, regardless of the dose of coagulant, at a fixed temperature of 23 °C (Figure 23). When the coagulant dose is set at 10 mL, the response surface plot for the interaction between temperature and pH shows that at any temperature, the COD removal has higher effectiveness around pH 10 (Figure 24). The sensitivity of COD removal to a slight change in pH may be due to the strong pH dependence of the number of active sites in the biocoagulant, which directly affects the removal rate of chemical compounds found in raw wastewater. In Figure 25, the center of the dose and coagulant temperature plot forms a kind of vertical/horizontal symmetry. More than 80% COD removal was obtained when the dose of the coagulant was set at 10 mL at pH 12. The studied biocoagulant was found to be effective in reducing solids and organics. The figures clearly indicate that the percentage of COD removal increases with increasing biocoagulant concentration.

### 3.5. Optical Study

Wastewater discoloration due to the coagulation treatment is shown in Figure 26, in terms of absorption reduction vs. coagulant dosage, for the three studied wavelengths. The lowest absorbance was noted at an LC dose of 12 mL, for which the absorption recorded at wavelengths of 620 and 436 nm had values of 0.043 and 0.023 Abs, respectively. At 525 nm, an LC dose of 6 mL produced the best reduction, with an absorbance of 0.057 Abs. These observations are in agreement with the results obtained by applying the RSM approach. The color removal based on the absorbance reductions at the maximum wavelengths was approximately 97.8%, 98.8% and 97.1% for the food wastewater at 620, 436 and 525 nm, respectively. Therefore, this variation in discoloration could be due to the absorption of the compounds in the treated wastewater [71].

The treatment of food wastewater with LC extract under optimal conditions modifies the appearance of the raw wastewater from a dark brown color (Figure 27a) to a significant discolored and limpid state (Figure 27b). During the coagulation treatment, floc formation clearly appears before the sedimentation phase. A period of 30 min was sufficient to provide complete floc settling and separation from the treated FW, without the addition of any flocculant.

## 4. Conclusions

This study unveiled another option for recovering and valorizing the deproteinization waste of chitin extraction from cuttlefish bone.

Analysis by SEM/EDX and XRD showed that the main constituent of cuttlefish bone was aragonite, one of the most common natural crystalline forms of calcium carbonate.

The presence of soluble protein fractions in the deproteinization liquid, in a concentration of 0.407 nM/mL, was proven by OPA assay for protein quantitation.

Analysis by SEM, ATR-FTIR, XRD and ^1^H-NMR proved that the deproteinization liquid extract contained soluble chitin forms, released from the cuttlefish bone during the deproteinization process; the concentration of chitin forms in the deproteinization liquid extract was 32 μg/mL.

Based on the zeta potential measurements of chitin fractions in alkaline media, these can produce coagulation in organic colloidal systems, such as food wastewater.

The coagulation test results indicate that the liquid extract resulting from CB deproteinization is effective in removing turbidity (99.68%), COD (82.92) and BOD_5_ (97.76%) when treating wastewater from the food processing industry at an alkaline pH (11) and a dose of 8 mL in 0.5 L of FW.

The wastewater treatment process was statistically modeled and optimized using a response surface methodology based on the Box–Behnken design. The quadratic models established for the removal of turbidity, COD and BOD_5_ show good predictability and reliability with regard to the experimental data.

The liquid extract resulted from cuttlefish bone deproteinization can be used as a biocoagulant for the efficient treatment of FWs. The proposed method is straightforward and cost-effective for highly polluted wastewater from food processing plants.

## Figures and Tables

**Figure 1 materials-15-02803-f001:**
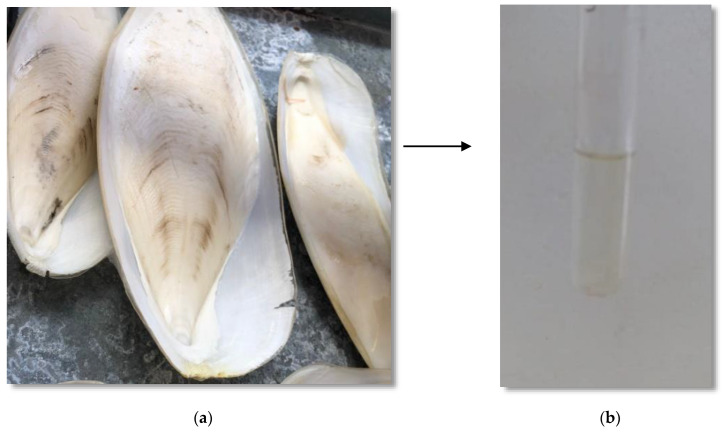
Transformation of cuttlefish bone into liquid extract after the deproteinization process: (**a**) *Sepia officinalis* bone and (**b**) Liquid extract from the deproteinization process.

**Figure 2 materials-15-02803-f002:**
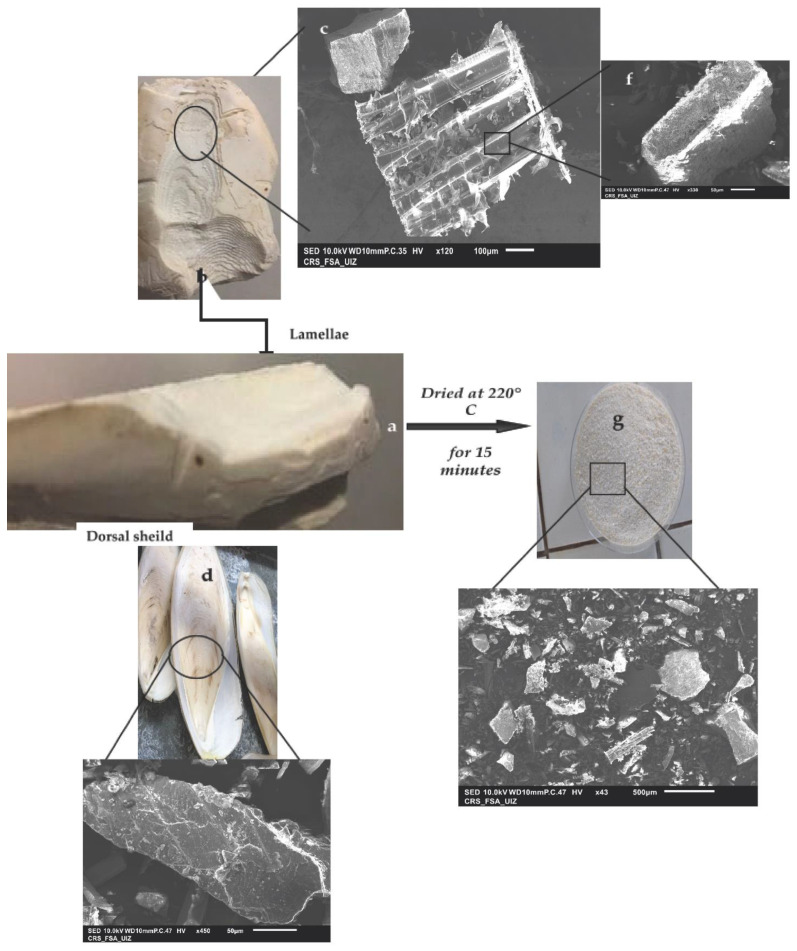
Enlarged detail of all parts of the CB using a Scanning Electron Microscope (SEM): (**a**) real photo of *sepia officinalis* bone, (**b**) real photo of *sepia officinalis* lamellae, (**c**) lamellae spacing analysis by SEM, (**d**) real photo of *sepia officinalis* dorsal shield; (**e**) a part of CB organic layer is shown, (**f**) enlarged detail of a lamellae analysis by SEM (**g**) global view of CB powder, (**h**) global view of CB powder by SEM.

**Figure 3 materials-15-02803-f003:**
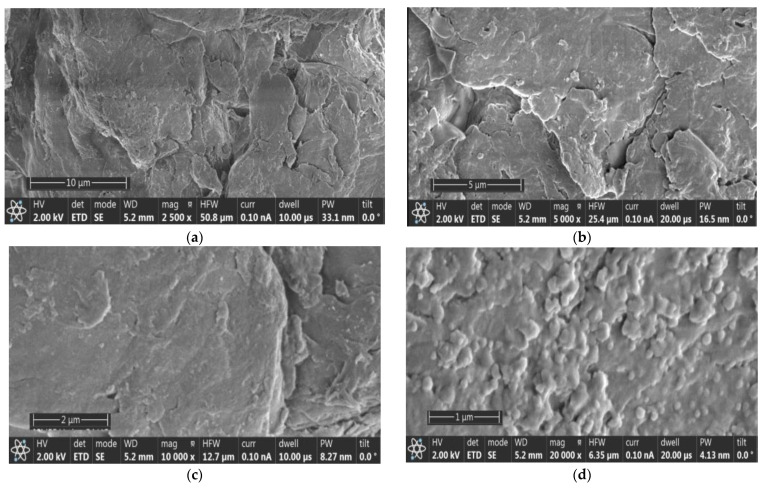
SEM images of chitin forms separated from the CB deproteinization liquid extract: (**a**) 2500× mag-10 µm scale; (**b**) 5000× mag-5 µm scale; (**c**) 10,000× mag-2 µm scale; (**d**) 20,000× mag-1 µm scale.

**Figure 4 materials-15-02803-f004:**
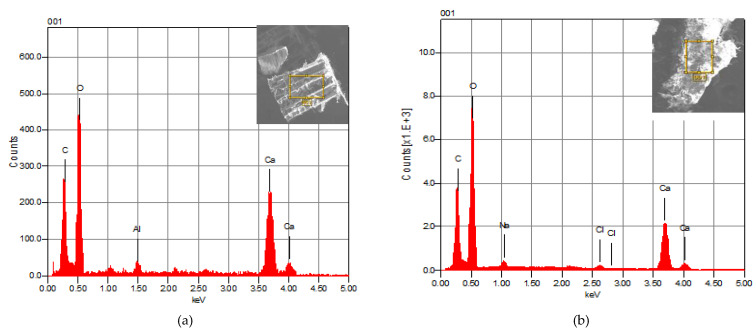
EDX spectra of CB powder (**a**): for 100 μm CB lamellae SEM image and (**b**): for 50 μm CB, dorsal shield SEM image.

**Figure 5 materials-15-02803-f005:**
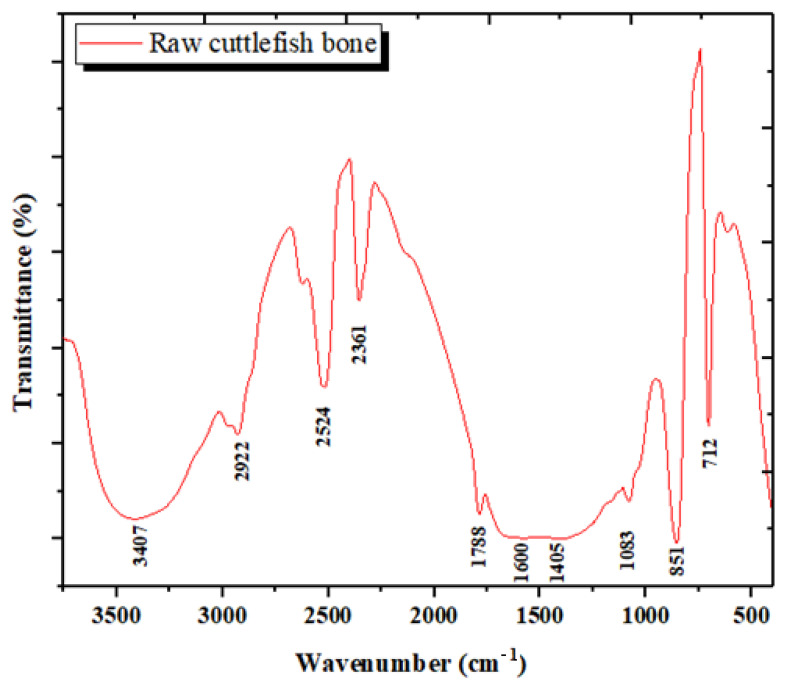
ATR FT-IR spectrum of raw CB powder.

**Figure 6 materials-15-02803-f006:**
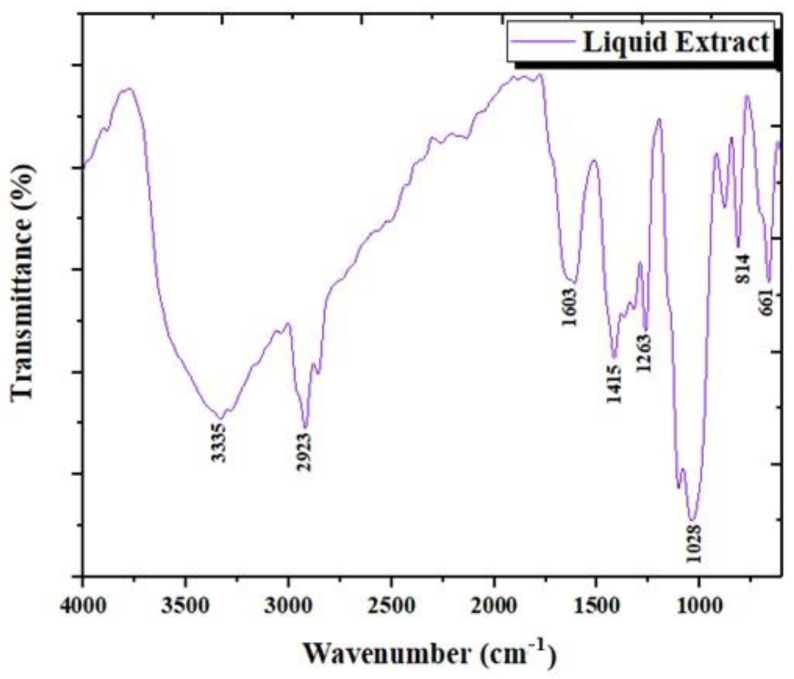
ATR FT-IR spectrum of CB deproteinization liquid extract (LC).

**Figure 7 materials-15-02803-f007:**
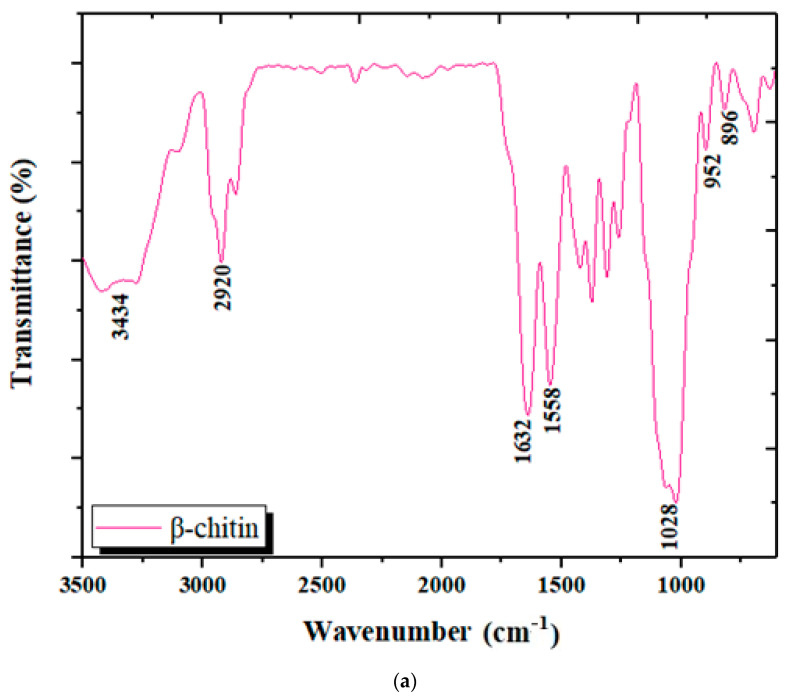
(**a**) The FT-IR spectrum of the chitin fraction separated from the cuttlefish bone deproteinization liquid extract; (**b**,**c**) deconvolution of two spectral domains of interest in the identification of chitin derivatives.

**Figure 8 materials-15-02803-f008:**
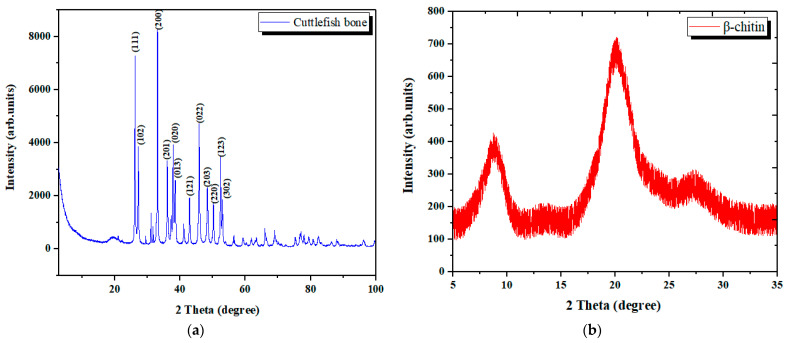
XRD spectra of cuttlefish bone materials: (**a**) XRD spectrum of raw cuttlefish bone powder; (**b**) XRD spectrum of chitin oligosaccharides isolated from the deproteinization liquid.

**Figure 9 materials-15-02803-f009:**
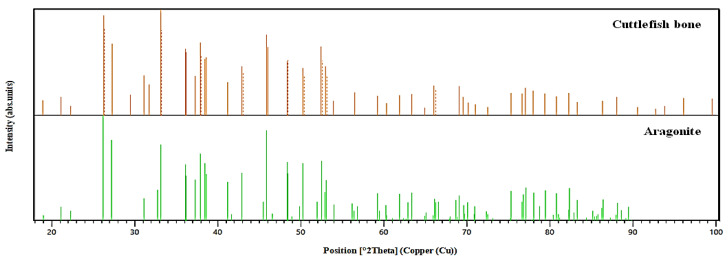
Comparison of XRD patterns of CB powder and aragonite.

**Figure 10 materials-15-02803-f010:**
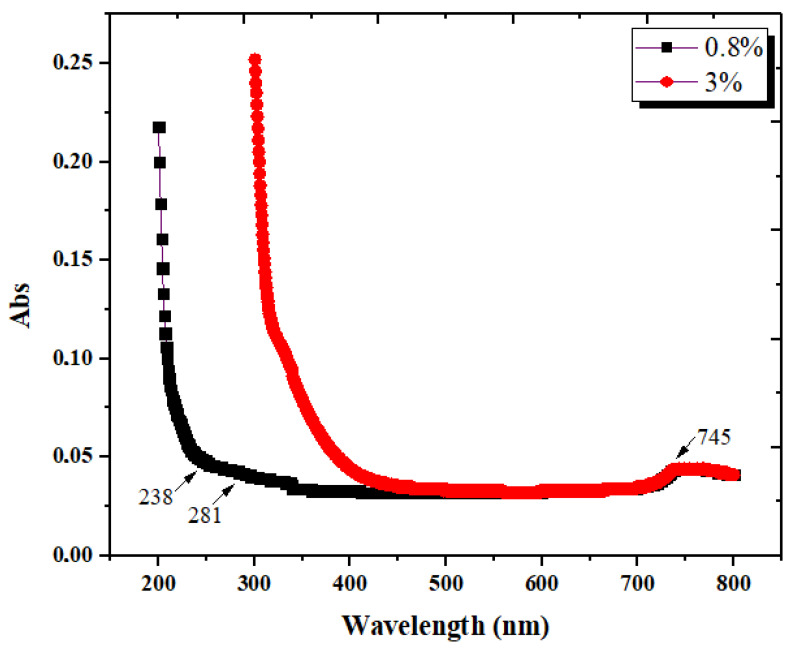
UV-Vis spectrum of the LC extract at two different concentrations.

**Figure 11 materials-15-02803-f011:**
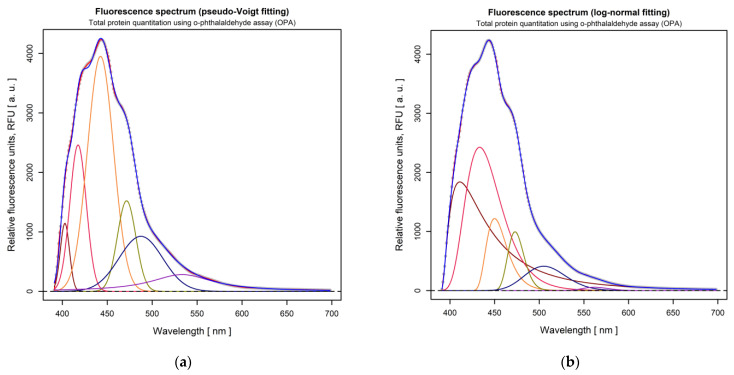
Total protein quantitation using OPA assay: (**a**) Fluorescence emission fitting by using Pseudo-Voigt; (**b**) log-normal distribution functions.

**Figure 12 materials-15-02803-f012:**
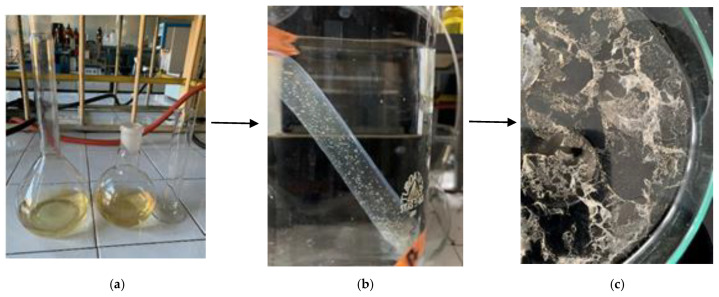
Separation of chitin fractions from the deproteinization liquid extract: (**a**) LC extract; (**b**) insolubilized pellets after dialysis in D_2_O and (**c**) purified chitin fractions for analysis.

**Figure 13 materials-15-02803-f013:**
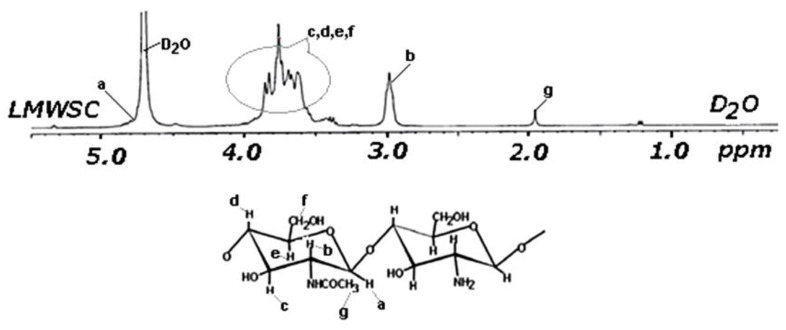
^1^H-NMR spectrum of low molecular weight water-soluble chitosan (LMWSC), adapted from [57].

**Figure 14 materials-15-02803-f014:**
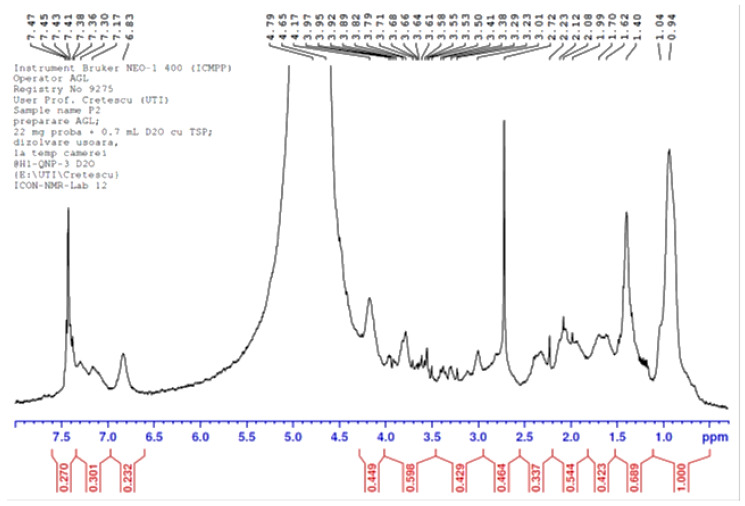
The enhanced-resolution ^1^H-NMR spectra of the chitin-based oligosaccharides separated from the deproteinization solution.

**Figure 15 materials-15-02803-f015:**
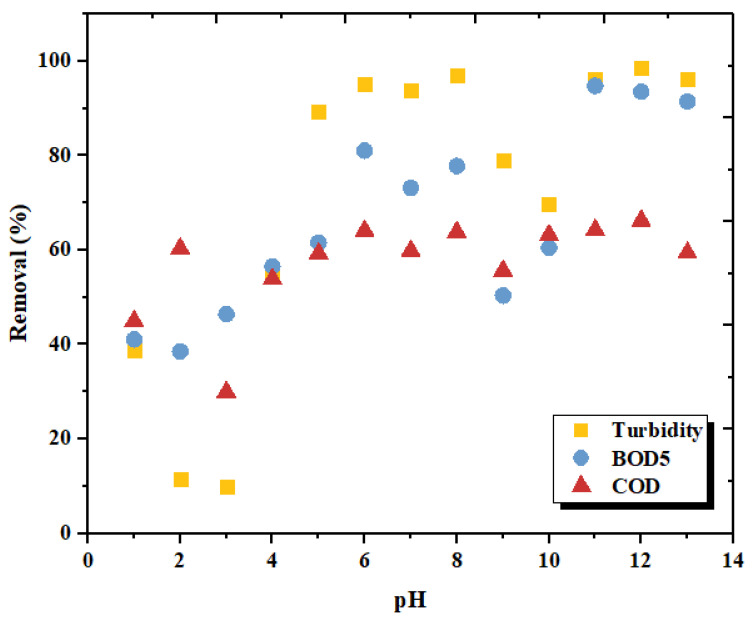
Effect of pH on turbidity, BOD_5_ and COD removal efficiency.

**Figure 16 materials-15-02803-f016:**
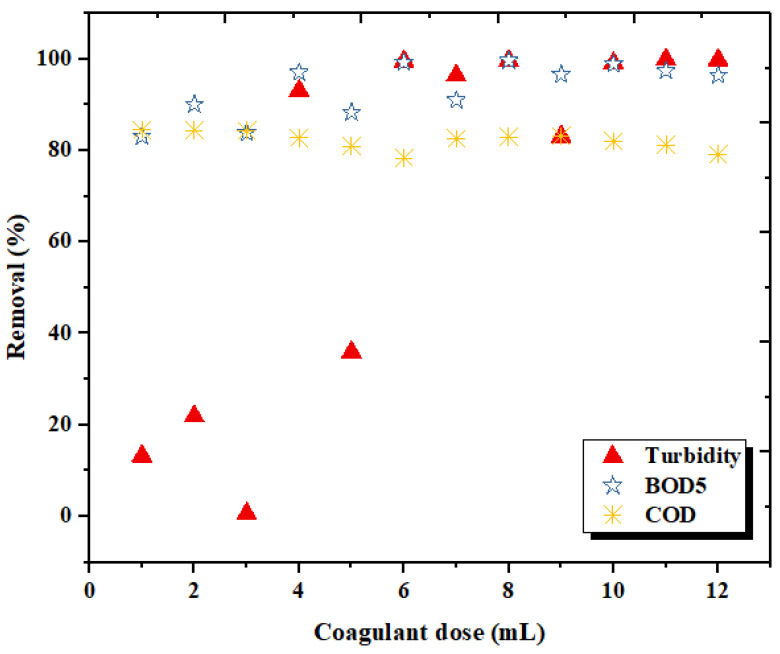
Effect of LC doses on turbidity, BOD_5_ and COD removal.

**Figure 17 materials-15-02803-f017:**
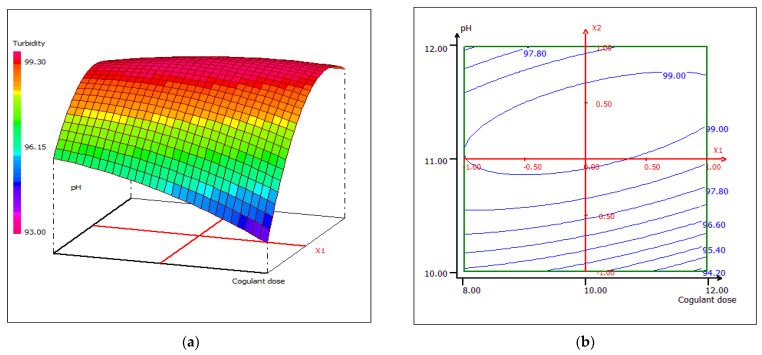
Three-dimensional (**a**) and two-dimensional (**b**) response surface plots for the turbidity of the treated food processing wastewater using β-chitin extract as a function of coagulant dose and pH, when the temperature was maintained at 23 °C.

**Figure 18 materials-15-02803-f018:**
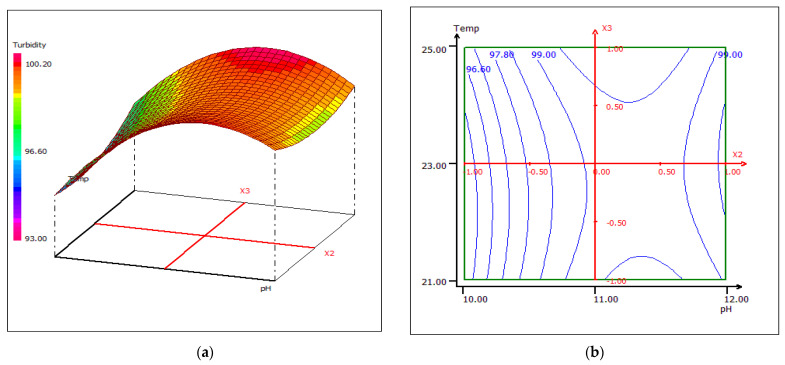
Three-dimensional (**a**) and two-dimensional (**b**) response surface plots for the turbidity of the treated food wastewater using β-chitin extract as a function of temperature and pH, when the coagulant dose was maintained at 10 mL.

**Figure 19 materials-15-02803-f019:**
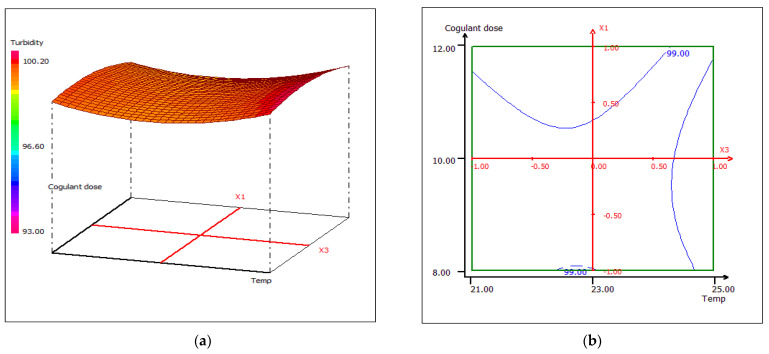
Three-dimensional (**a**) and two-dimensional (**b**) response surface plots for the turbidity of the treated food wastewater using β-chitin extract as a function of coagulant dose and temperature when the pH was maintained at 11.

**Figure 20 materials-15-02803-f020:**
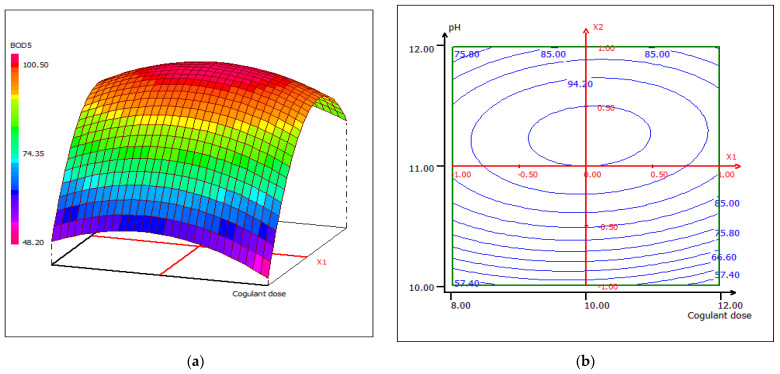
Three-dimensional (**a**) and two-dimensional (**b**) response surface plots for the BOD_5_ of the treated food wastewater using β-chitin extract as a function of coagulant dose and pH, when the temperature was maintained at 23°C.

**Figure 21 materials-15-02803-f021:**
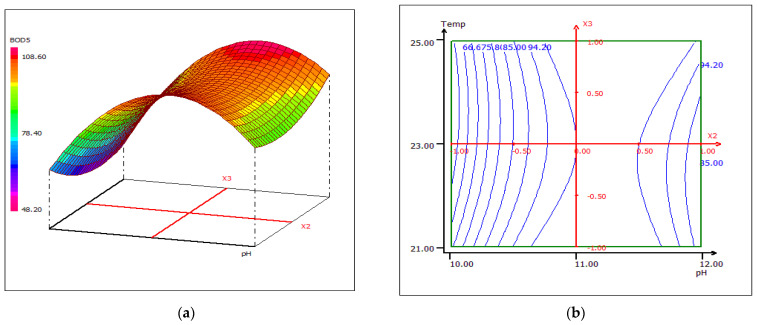
Three-dimensional (**a**) and two-dimensional (**b**) response surface plots for the BOD5 of the treated food wastewater using β-chitin extract as a function of temperature and pH, when the coagulant dose was maintained at 10 mL.

**Figure 22 materials-15-02803-f022:**
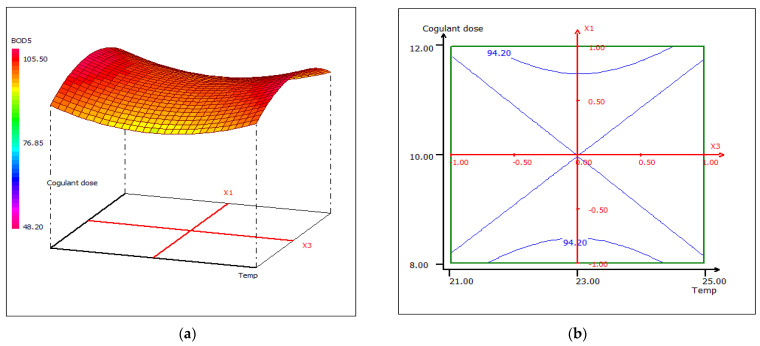
Three-dimensional (**a**) and two-dimensional (**b**) response surface plots for the BOD5 of the treated food wastewater using β-chitin extract as a function of coagulant dose and temperature, when the pH was maintained at 11.

**Figure 23 materials-15-02803-f023:**
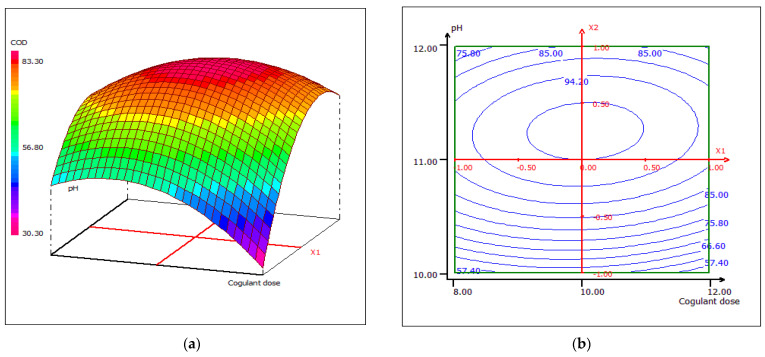
Three-dimensional (**a**) andtwo-dimensional (**b**) response surface plots for the COD of the treated food wastewater using β-chitin extract as a function of coagulant dose and pH, when temperature was maintained at 23 °C.

**Figure 24 materials-15-02803-f024:**
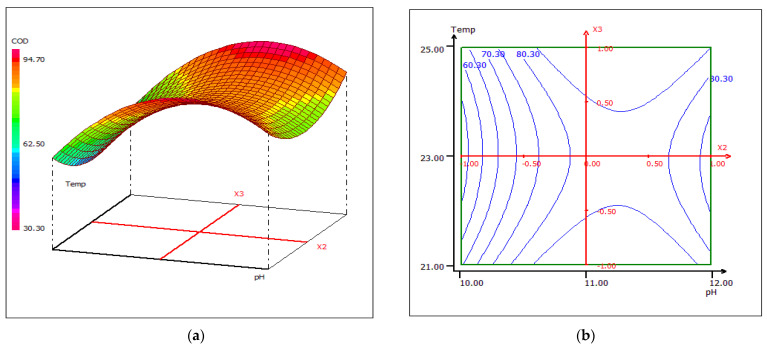
Three-dimensional (**a**) and two-dimensional (**b**) response surface plots for the COD of the treated food wastewater as a function of temperature and pH, when the coagulant dose was maintained at 10 mL.

**Figure 25 materials-15-02803-f025:**
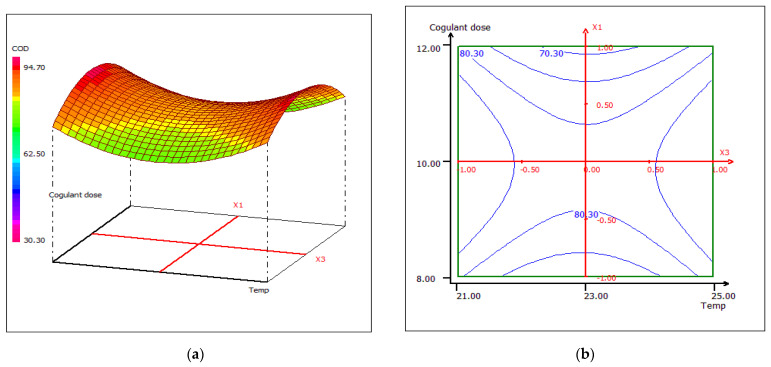
Three-dimensional (**a**) and two-dimensional (**b**) response surface plots for the COD of the treated food wastewater using β-chitin extract as a function of coagulant dose and temperature, when the pH was maintained at 11.

**Figure 26 materials-15-02803-f026:**
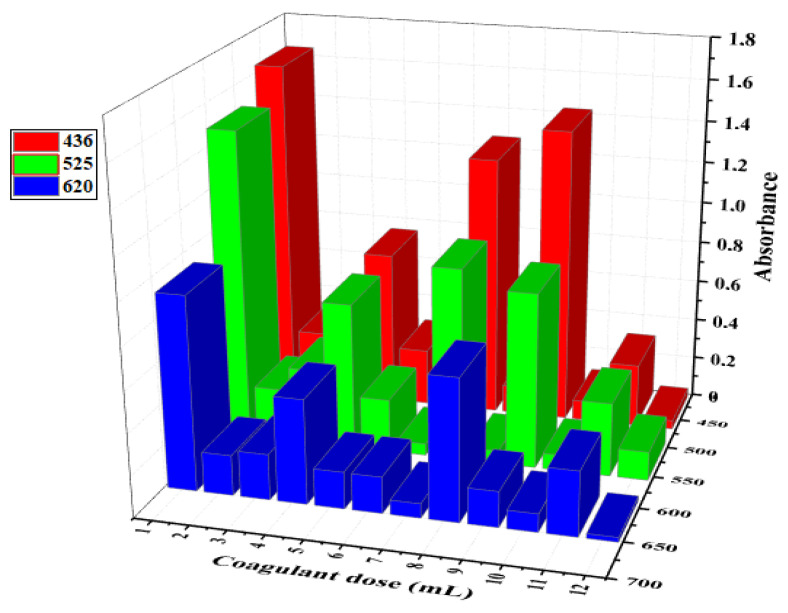
UV-Vis absorbance of treated FW vs. coagulant dose at three different wavelengths: 436, 525 and 620 nm.

**Figure 27 materials-15-02803-f027:**
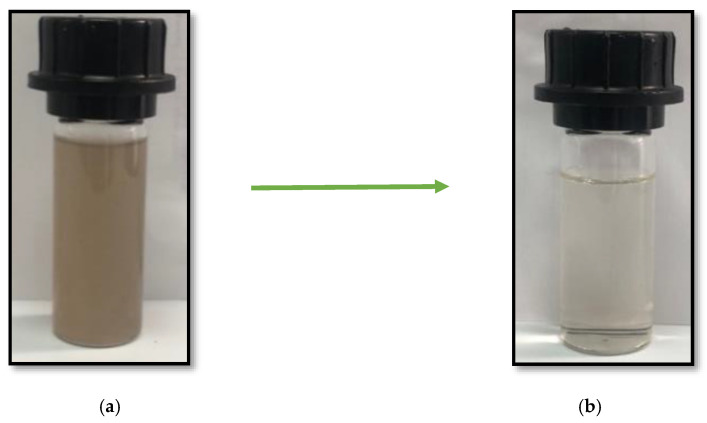
Wastewater discoloration produced by the coagulation treatment: (**a**) Raw FW; (**b**) FW after coagulation treatment using the LC biocoagulant.

**Table 1 materials-15-02803-t001:** Experimental design matrix for FW treatment using β-chitin extract.

No Exp	Experimental Parameters			Responses		
	X_1_ Coagulant Dose (mL)	X_2_ Initial pH	X_3_ Temperature (°C)	Y_1_ Turbidity Removal (%)	Y_2_ BOD_5_ Removal (%)	Y_3_ COD Removal (%)
1	8.00	10.00	23.00	96.00	52.64	50.35
2	12.00	10.00	23.00	93.00	48.17	30.25
3	8.00	12.00	23.00	98.00	75.75	53.50
4	12.00	12.00	23.00	99.07	80.32	68.96
5	8.00	11.00	21.00	99.12	97.89	80.91
6	12.00	11.00	21.00	98.90	97.58	79.90
7	8.00	11.00	25.00	99.68	97.76	82.92
8	12.00	11.00	25.00	99.85	96.46	79.09
9	10.00	10.00	21.00	96.44	70.63	65.34
10	10.00	12.00	21.00	98.32	86.42	82.18
11	10.00	10.00	25.00	97.43	61.30	62.75
12	10.00	12.00	25.00	98.00	97.11	84.73
13	10.00	11.00	23.00	99.13	98.80	81.94
14	10.00	11.00	23.00	99.13	98.80	81.94
15	10.00	11.00	23.00	99.13	98.80	81.94

**Table 2 materials-15-02803-t002:** Experimental range and levels of independent factors.

Coded Variables	Factor	Coded Level		
		−1	0	+1
X_1_	Coagulant dosage (mL/0.5 L)	8	10	12
X_2_	Initial pH	10	11	12
X_3_	Temperature (°C)	21	23	25

**Table 3 materials-15-02803-t003:** Physicochemical characterization of FW [25].

Parameter	Value			Moroccan Standard Limit of Direct Discharge
	Min	Mean	Max	
Temperature	12	18.5	25	30 °C
pH	6.62	7.06	7.50	5.5–9.5
Conductivity	9000	15,500	22,000	2700 µS/cm
Turbidity	850	925	>1000	-
BOD_5_	1700	2345.5	2990.9	100 mg O_2_/L
COD	2380	3735	5090	500 mg O_2_/L

**Table 4 materials-15-02803-t004:** The pH dependence of zeta potential of chitin oligosaccharides.

pH of colloidal suspension	7.0	8.0	9.0	10.0	11.0
Measured zeta potential (mV)	+10.4	+2.07	−3.97	−7.12	−10.3

**Table 5 materials-15-02803-t005:** Statistical analysis of the regression coefficients estimated due to the FW variation.

Parameter	Standard Deviation	R^2^	R^2^ Adjusted	R^2^ Predicted
Turbidity	0.946	0.901	0.722	N.D
BOD_5_	0.760	0.999	0.998	0.990
COD	0.520	1.00	0.999	0.994

**Table 6 materials-15-02803-t006:** Analysis of variance (NemrodW) for the FW treatment models.

Coefficient		b_0_	b_1_	b_2_	b_3_	b_1–1_	b_2–2_	b_3–3_	b_1–2_	b_1–3_	b_2–3_
**Value**	**Turbidity**	99.130	−248	1.315	0.273	−0.386	−2.226	0.644	1.017	0.097	−0.328
	**BOD_5_ removal**	98.800	−0.189	13.357	0.014	−8.011	−26.569	6.634	2.248	−0.248	5.005
	**COD removal**	81.940	−1.185	10.085	0.145	−12.110	−19.065	10.875	8.890	−0.705	1.285
**F. inflation**	**Turbidity**	-	1.00	1.00	1.01	1.01	1.01	1.00	1.00	1.00	1.00
	**BOD_5_ removal**	-	1.00	1.00	1.00	1.01	1.01	1.01	1.00	1.00	1.00
	**COD removal**	-	1.00	1.00	1.00	1.01	1.01	1.01	1.00	1.00	1.00
**Standard** **deviation**	**Turbidity**	0.546	0.335	0.335	0.335	0.493	0.493	0.493	0.473	0.473	0.473
	**BOD_5_ removal**	0.439	0.269	0.269	0.269	0.396	0.396	0.396	0.380	0.380	0.380
	**COD removal**	0.300	0.184	0.184	0.184	0.271	0.271	0.271	0.260	0.260	0.260
**t exp**	**Turbidity**	181.43	−0.74	3.93	0.81	−0.78	−4.52	1.31	2.15	0.21	−0.69
	**BOD_5_ removal**	225.03	−0.70	49.68	0.05	−20.24	−67.13	16.76	5.94	−0.65	13.16
	**COD removal**	272.89	−6.44	54.85	0.79	−44.74	−70.44	40.18	34.19	−2.71	4.94
**Significance (%)**	**Turbidity**	<0.01 ***	49.3	1.11 *	45.2	46.8	0.628 **	24.8	8.4	84.5	52.0
	**BOD_5_ removal**	<0.01 ***	51.4	<0.01 ***	96.1	<0.01 ***	<0.01 ***	<0.01 ***	0.193 **	54.4	<0.01 ***
	**COD removal**	<0.01 ***	0.134 **	<0.01 ***	46.6	<0.01 ***	<0.01 ***	<0.01 ***	<0.01 ***	4.22 *	0.432 **
**Sources of** **Variation**				**Regression**		**Residual**			**Total**		
**Sum of squares**	**Turbidity**			40.572		4.478			45.050		
	**BOD_5_ removal**			4.59662 × 10^3^		2.891			4.59951 × 10^3^		
	**COD removal**			3.55346 × 10^3^		1.352			3.55481 × 10^3^		
**Degree of freedom**	**Turbidity**			9		5			14		
	**BOD_5_ removal**			9		5			14		
	**COD removal**			9		5			14		
**Mean square**	**Turbidity**			4.5080		0.8956			-		
	**BOD_5_ removal**			5.10736 × 10^2^		5.78295 × 10^2^			-		
	**COD removal**			3.94829 × 10^2^		2.70480 × 10^−1^			-		
**Ratio**	**Turbidity**			5.033		-			-		
	**BOD_5_ removal**			883.174		-			-		
	**COD removal**			1459.734		-			-		
**Significance (%)**	**Turbidity**			4.49 *		-			-		
	**BOD_5_ removal**			<0.01 ***		-			-		
	**COD removal**			<0.01 ***		-			-		

*** Extremely significant; ** Very significant; * Significant.

## Data Availability

Data is contained within the article.

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
