# Peer review of "Valorization of β-Chitin Extraction Byproduct from Cuttlefish Bone and Its Application in Food Wastewater Treatment"

_materials, 2022, doi:10.3390/ma15082803_

Round 1

Reviewer 1 Report

In the Abstract, the authors should much more focus on elaborating the main results acquired from this work rather than describing what kind of experiments/analysis done in this work.

In Figure 2, several figures were not clear. I recommend that the authors could provide higher resolution images.

A proof reading by a native English speaker should be conducted to improve both language and organization quality.

Some minor mistake should be avoided. For example, in Page 15: R square value – “2” should be in superscript.

I recommend that the conclusion section should be amended to better highlight the main results from this work. Now I feel that only partial results are presented in the conclusion section. (for example, the authors conducted various physicochemical characterization; however, no any characterization results were shown in the conclusion section)

Author Response

Please find the answers in the attached file

Reviewer 2 Report

The present work outlines the extraction of β-chitin from Cuttlefish
bone and its application for food wastewater treatment. The work sounds good with a little originality. The results are promising. Indeed, several issues should be handled carefully.

  1. The introduction section should visualize the potential of β-chitin extracted from Cuttlefish bone within the wastewater treatment process.
  2. SEM images are not clear, authors are suggested to provide clear images with different magnification.
  3. In page 7 "In addition, the adsorption bands at 600-700 cm-1 may also be due to the chitin content of the CB powder" It is not adsorption bands, instead, IR peaks should be used.
  4. Page 7, " A broad peak at 3422cm-1 was ascribed to C-H stretching vibration mode of the hydroxyl group" should be O-H not C-H.
  5.  The protein band is visible around 2000 cm-1 for the secondary amide bond N-H". This interpretation is incorrect, there is no IR peaks for N-H group at this position.
  6. It is better to overlay the XRD results together with aragonite pattern in the same chart for better comparison.
  7. 1HNMR is needed to confirm the identity of ß-chitin.
  8. Figure 10 depicts the removal efficiency as a function of pH. Results shows that better treatment was achieved both at pH=6 and 11. Authors interpreted the performance under alkaline conditions only. It would be better if they provide additional explanation for the better performance at pH=6.
  9. English language should be checked,  for example "tone" in page 1.

Author Response

(The authors gave the same response as above.)

Reviewer 3 Report

In this paper, organic substances from cuttlefish bone were used to remove impurities in food industrial wastewater by the adsorption method. The optimal values of adsorbent dosage, pH value, and temperature were determined by response surface methodology. The adsorption material was cheap and novel. It is recommended to provide the specific surface areas of the adsorbent. Besides, in the adsorption experiment, the Zeta potential test data can be used to observe the charge charges of the particles under different adsorption conditions.  It is necessary to continue to measure the cyclic removal performance of the adsorbent under the optimal adsorption conditions after one desorption, and further explore the reuse value of the material. Some pictures are not clear enough.

Author Response

(The authors gave the same response as above.)

Round 2

Reviewer 2 Report

I appreciate the authors' efforts to improve the work's quality. However, I can not recommend publication at current stage. Reasons are below.

  1. XRD, SEM and IR characterizations should be performed for purified B-chitin sample not CB powder.
  2. Much efforts should be devoted to improve the organization, interpretation and discussion of the results. The main results are hardly detectable. 
  3. The Zeta potential results should correlated to the observed performance clearly.

Author Response

We appreciate the reviewer’s efforts and their time for carefully reading, editing, and commenting on our manuscript. We are very thankful for their critical comments and constructive suggestions, which helped us to improve the manuscript quality. We have revised our manuscript based on the reviewer’s comments and gave detailed answers/justifications to their comments/suggestions/queries. The responses to the reviewer’s comments are given in the attached file! 

All the changes done in the manuscript during this revision are shown directly in the revised text highlighted in yellow color (black text-first revision round) and (red text-second revision round)

Reviewer 3 Report

Accept in present form

Author Response

We appreciate the reviewer’s efforts and their time for carefully reading, editing, and commenting on our manuscript. We are very thankful for their critical comments and constructive suggestions, which helped us to improve the manuscript quality. We have revised our manuscript based on the reviewer’s comments and gave detailed answers/justifications to their comments/suggestions/queries. The responses to the reviewer’s comments are given in the attached file.

All changes done in the manuscript during this revision are shown directly in the revised text highlighted in yellow color (black text-first revision round) and (red text-second revision round)
